# Electrodischarge Methods of Shaping the Cutting Ability of Superhard Grinding Wheels

**DOI:** 10.3390/ma14226773

**Published:** 2021-11-10

**Authors:** Marcin Gołąbczak, Robert Święcik, Andrzej Gołąbczak, Dariusz Kaczmarek, Ryszard Dębkowski, Barbara Tomczyk

**Affiliations:** 1Department of Production Engineering, Institute of Machine Tools and Production Engineering, Lodz University of Technology, Stefanowskiego Str. 1/15, 90-924 Łódź, Poland; robert.swiecik@p.lodz.pl (R.Ś.); cezahr@wp.pl (D.K.); ryszard.debkowski@p.lodz.pl (R.D.); 2Department of Management Engineering, Institute of Social and Technical Sciences, State Vocational University in Włocławek, 3 Maja Str. 17, 87-800 Włocławek, Poland; andrzej.golabczak@puz.wloclawek.pl; 3Department of Mechanics and Building Structures, Institute of Civil Engineering, University of Life Sciences, Nowoursynowska Str. 166, 02-787 Warszawa, Poland; barbara_tomczyk@sggw.edu.pl

**Keywords:** superhard grinding wheels, electrodischarge dressing, cutting ability

## Abstract

In the paper, the influence of the electrodischarge dressing methods of superhard grinding wheels on shaping their cutting ability are presented. The results of research concerning the influence of dressing conditions using a stationary electrode, rotating electrode and segmental tool electrode on shaping the cutting ability of the superhard grinding wheels are reported. The cutting ability of superhard grinding wheels is assessed using an external tester made of titanium alloy Ti-6Al-4V (with a thermocouple) to determine the grinding temperature and the relative volumetric grinding efficiency of the tool. The results of the research reveal the diversified usefulness of the analyzed methods. At the end of the article application conclusions concerning the adaptation of developed methods of electrodischarge dressing in the industry are formulated.

## 1. Introduction

Abrasive grinding is still a key method of efficiently and accurately shaping workpieces, especially those made of difficult-to-machine construction materials with high hardness and strength, e.g., titanium, magnesium alloys, nickel and cobalt-based superalloys, Inconel, etc. [1,2,3]. The increasing requirements for modern manufacturing processes in the field of: dimensional accuracy, surface roughness of the surface layer condition, productivity and flexibility of production, resulted in the development of new, innovative varieties and methods of grinding and abrasive tools.

The new, innovative grinding methods include: high-speed grinding (HSG) [4,5], high efficiency deep grinding (HEDG) [6,7,8], grinding with reduced grinding wheel contact (Quickpoint) [9,10], and hybrid grinding: electrochemical grinding (AECG) [11,12,13,14], electrolytic in-process grinding—ultra-precision grinding (ELID) [15,16], electrodischarge grinding (AEDG) [17,18,19,20]. These grinding methods mainly use modern grinding wheels made of superhard abrasives (natural diamond—ND, synthetic diamond—SD or cubic boron nitride—cBN), bonded with a metal bond.

The rational and economic use of superhard grinding wheels requires the use of unconventional, erosive dressing methods, ensuring effective shaping macro- and microgeometry of the cutting surface of the grinding wheel (CSGW), and thus the cutting ability of the grinding wheel. The article presents the results of research on the evaluation of the cutting ability of grinding wheels made of diamond and cBN abrasives in the process of electrodischarge dressing with a stationary electrode, rotating electrode and segment electrode. To assess the cutting ability of grinding wheels, a two-parameter method of grinding the external tester was used, which allowed to determine: the grinding temperature—*Θ_t_* and the relative volumetric grinding efficiency—*Q_v_* [21].

## 2. Methods of Erosive Shaping of the Cutting Ability of Superhard Grinding Wheels

The analysis of source publications and own research indicates that conventional methods of dressing superhard grinding wheels with a metal bond (e.g., dressing with diamond blade dressers, rotating dressers, whetstones and abrasives, chipping with a hard roller) only to a limited extent allow for shaping the macro- and microgeometry of the CSGW [22,23,24]. The limitations of these methods result mainly from the specific physical and mechanical properties of superhard grinding wheels, especially the high hardness of the abrasive and the strength of the metal bond. In order to counteract the occurrence of the grinding wheels loading, the grinding wheels can be impregnated with molybdenum compounds (MoS_2_) [25].

Rational and economic shaping of the CSGW macro- and microgeometry of metalbonded superhard grinding wheels, and thus shaping their cutting abilities, is ensured by erosive dressing methods. The basic methods of erosive dressing of superhard grinding wheels include: high-pressure abrasive water jet dressing [24,26], laser dressing [24,27,28], electrochemical dressing [24,29,30,31,32], electrolytic in-process dressing ELID [33,34], electrochemical dressing with foil electrodes [35], electrodischarge dressing [24,36,37,38,39] and hybrid electrodischarge-electrochemical dressing [40,41,42].

The article presents the results of research on shaping the cutting ability of diamond and cBN superhard grinding wheels in the process of electrodischarge dressing with a stationary, rotating and segment electrode. The advantages of these dressing methods are: better efficiency of the dressing process, low environmental impact and possibility of their application in industrial conditions, compared to other methods, e.g., electrochemical or laser dressing.

## 3. Materials and Methods

The aim of the work is to investigate the influence of the conditions of electrodischarge dressing of metal-bonded superhard grinding wheels on the shaping of their cutting abilities. Three types of dressing were used to shape the cutting ability of grinding wheels, i.e., dressing with a stationary electrode, a rotating electrode and a segmental electrode. To assess the cutting ability of superhard grinding wheels after electrodischarge dressing, a two-parameter method of external tester grinding was used.

This method was used to determine the grinding temperature—*Θ_t_* and the relative volumetric grinding efficiency—*Q_v_* [21]. The grinding temperature—*Θ_t_* [°C] is determined with constant infeed speed of the tester to the grinding wheel—*v_fw_* [µm s^−1^], however, the relative volumetric grinding efficiency—*Q_v_* is determined with constant infeed force of the tester to the grinding wheel—*F* [N] and is defined by the following relationship:(1)$$Q_{v}=\frac{V_{w}}{t}\;\left[\frac{{\mathrm{mm}}^{3}}{s}\right]$$
where: *V_w_*—volume of grinded tester [mm^3^], *t*—tester grinding time [s].

The relative volumetric grinding efficiency—*Q_v_* value was calculated according to the computer program based on the recorded grinding parameters of the tester, i.e., tester grinding time—*t* and the linear loss of the tester—Δ*h_t_*. The volume of grinded tester *V_w_* was calculated according to the following relationship:(2)$$V_{w}=\frac{\pi d_{t}{}^{2}}{4}\Delta h_{t}\;\left[{\mathrm{mm}}^{3}\right]$$
where: *d_t_*—tester diameter [mm], Δ*h_t_*—linear loss of the tester [mm].

The grinding wheel dressing tests were carried out on an electrochemical plane grinding machine ECBT8, equipped with a Marcosta GMP75 spark pulse generator and control and measurement systems for dressing superhard grinding wheels (Figure 1). The basic parameters of the grinder and grinding wheels used in the tests are presented in Table 1.

The grinder was equipped with a device for a two-parameter assessment of the cutting ability of grinding wheels by means of external tester grinding and three systems for electrodischarge dressing of superhard grinding wheels with a stationary, rotating and segment electrode. The research also used the Hitachi S-3500 N scanning microscope, the Kistler 9257B piezoelectric dynamometer and the PGM-1C IOS Kraków profilometer.

### 3.1. System of Electrodischarge Dressing of Grinding Wheels with a Stationary Electrode

Electrodischarge dressing of grinding wheels with a stationary electrode is the simplest method of dressing. However, it requires isolating the grinder spindle from the body and a commutator-type voltage supply to the grinding wheel spindle (Figure 2).

In this dressing method, a rectangular-wall graphite block (E28S) with dimensions of 120 × 25 × 35 mm was used as the working electrode, mounted in a vice on the grinding table, to which the negative voltage pole from the electro-spark discharge generator was connected.

During the electrodischarge dressing of superhard grinding wheels with this system, the dielectric was fed with nozzles to the working gap formed between the rotating grinding wheel and the electrode. Distilled water was used as the dielectric liquid. The grinding table with the stationary electrode made a reciprocating movement with the speed *v_fe_*. After each of the table turns, the grinding wheel was moved to the working electrode, keeping the working gap of approximately 0.05 mm.

### 3.2. System of Electrodischarge Dressing of Grinding Wheels with a Rotating Electrode

The system of electrodischarge dressing of grinding wheels with a rotating electrode is structurally more complex, because it requires not only insulation of the grinding wheel spindle but also an additional device with a drive for the rotating electrode and a commutator-driven voltage supply to the grinding wheel and the rotating electrode (Figure 3).

A graphite (E28S) disc with a diameter of 70 mm and a width of 30 mm was used as the working electrode. A pneumatic motor was used to drive the rotating electrode. The working electrode made a rotation in the concurrent direction with respect to the grinding wheel. The dressing process was carried out at a constant rotational speed of the rotating electrode (*n_e_* = 750 rpm), which was determined on the basis of the test results. A copper graphite brush commutator assembly was used to connect the negative pole of the voltage from the electro-spark generator to the rotating electrode, in contact with the rotating electrode mounting sleeve. Distilled water was used as the dielectric liquid. In the process of electrodischarge dressing of superhard grinding wheels, the dielectric was also fed with nozzles to the inter-electrode gap (0.05 mm) between the rotating grinding wheel and the co-rotating electrode.

### 3.3. System of Electrodischarge Dressing of Grinding Wheels with a Segmental Electrode

The system of electrodischarge dressing with a segmental electrode developed at the Lodz University of Technology has the advantage, compared to the previous two, that it does not require the spindle of the grinder to be insulated (Figure 4).

In this method of dressing of grinding wheels, the voltage from the spark discharge generator is applied directly to the segmental electrode. The electro-spark discharges, on the other hand, take place in the electrode gap between the metal binder of the grinding wheel and the electrode segments. In the process of electrodischarge dressing of superhard grinding wheels, the grinding table with a batten electrode made a reciprocating movement with longitudinal feed speed of the electrode—*v_fe_*. The grinding wheel was moved to the bar electrode after each of the turns of the table with a unit infeed (single infeed of the grinding wheel to the working electrode in dressing tests).

### 3.4. A Device for Assessing the Cutting Ability of Grinding Wheels with the Use of a Two-Parameter Method of Grinding the External Tester

The two-parameter method of grinding the external tester, developed at the Lodz University of Technology, consists of a several-second grinding test of a special tester, located outside the grinding zone of workpieces. Inside the grinded tester there is a K-type sheathed thermocouple. The indicators for assessing the grinding ability of the grinding wheels are recorded during the grinding test: the linear loss of the grinded tester moved to the grinding wheel at a constant longitudinal feed speed of the electrode—*v_fe_* and the grinding temperature—*Θ_t_* assessed by the tester which moved to the grinding wheel with a constant infeed force of the tester to the grinding wheel—*F*. The value of the infeed speed of the tester to the grinding wheel—*v_fw_* and the tester infeed force—*F* were determined on the basis of previous grinding tests, assuming, respectively: *v_fw_* = 24 µm/s, *F* = 2 N [21,30]. The view of the device for assessing the cutting ability of the grinding wheels using the two-parameter method of grinding the external tester mounted on the grinder is shown in Figure 5, while the diagram of its construction is shown in Figure 6.

The tester—2 (Ti-6Al-4V titanium alloy sample—5 mm in diameter with a *K*-type sheathed thermocouple—0.5 mm in diameter, mounted inside it), is connected to a force gauge—6 are attached to a roller bearing slide—7, ensuring precise infeed to the grinding wheel. The slide feed with the tester and dynamometer is carried out through the stepper motor—8, toothed gear—9 and helical-ball gear—15. During grinding of the tester, measurement signals from the dynamometer and the sheathed thermocouple are recorded in the computer. The measurement cycle for determining the *Q_v_* and *Θ_t_* parameters is controlled by a computer equipped with a measuring card and runs in two stages:The first stage includes: controlling the speed of the tester infeed to the grinding wheel (ensuring a constant grinding force) registration of the tester linear loss and determination of the *Q_v_* parameter value, the tester infeed speed during grinding is then regulated in the feedback path: force sensor, amplifier, stepper motor computer and driver;The second stage includes: controlling the tester infeed to the grinding wheel at a constant speed, recording the tester temperature and determining the average temperature *Θ_t_*; the constant speed of the tester infeed to the grinding wheel required in this measuring cycle is provided by the stepper motor control system, which implements the tester infeed according to the developed computer program using pulses generated by the computer clock.

## 4. Results and Discussion

The presented research results concern the assessment of the influence of the conditions of electrodischarge dressing of superhard grinding wheels using three methods, i.e., with a stationary, rotating and segment electrode, on shaping their cutting surface abilities. The scope of the research included tests of electrodischarge dressing using these methods of diamond and cBN grinding wheels. The variable values in these methods of dressing grinding wheels were the power of the electro-spark discharge generator (P1 ÷ P3) and the selected technological parameters in these processes. Variable power values of the electro-spark discharge generator were used: P1 (*U* = 100 V, *I* = 12 A), P2 (*U* = 200 V, *I* = 25 A), P3 (*U* = 300 V, *I* = 50 A). The constant values of the electro-spark discharge generator were: pulse duration *t_on_* = 500 µs and the time of breaks between discharges *t_off_* = 250 µs. To assess the cutting ability of grinding wheels in these electrodischarge methods, a two-parameter method of grinding the external tester was used, which was determined: the grinding temperature—*Θ_t_* and the relative grinding efficiency—*Q_v_*. The values of *Θ_t_* and *Q_v_* were established as the arithmetic mean of 5 measurement tests. 

### 4.1. Electrodischarge Dressing of Grinding Wheels with a Stationary Electrode

The tests included three series of electrodischarge dressing tests of grinding wheels with diamond and cBN abrasives using a stationary electrode, carried out with the following power settings of the electro-spark discharge generator: P1 (*U* = 100 V, *I* = 12 A); P2 (*U* = 200 V, *I* = 25 A); P3 (*U* = 300 V, *I* = 50 A). In individual dressing series, the unit infeed of the grinding wheel to the stationary electrode was differentiated, amounting to: series I—0.005 mm, series II—0.010 mm and series III—0.015 mm. The value of the total infeed of the grinding wheel to the electrode was 0.1 mm, while the dressing time in subsequent unit strokes was 60 s. The scope of the research included determining the effect of the power of the spark generator pulse generator and the unit infeed of the grinding wheel to the electrode on the grinding temperature—*Θ_t_* and relative grinding efficiency—*Q_v_*.

#### 4.1.1. Dressing Tests of Diamond Grinding Wheel

The results of the research concerning the assessment of the influence of the power of the electro-spark discharge generator and the unit infeed—*a_j_* of the diamond grinding wheel to the stationary electrode on the grinding temperature—*Θ_t_* and the relative grinding efficiency—*Q_v_* are presented in Figure 7.

The analysis of the test results of the dressing process of the diamond grinding wheel using stationary electrode showed that the increase of the power of the electro-spark discharge generator has a positive effect on the reduction of the grinding temperature—*Θ_t_* and the increase of the relative grinding efficiency—*Q_v_* (Figure 7). In particular dressing series, a reduction of the grinding temperature—*Θ_t_* was achieved, amounting to: in series I—5% ÷ 11%, in series II—4% ÷ 11% and in series III—5% ÷ 11%, and an increase of the relative grinding efficiency amounting to: in series I—3 % ÷ 8%, in series II—4% ÷ 8% and in series III—4% ÷ 7%. Increasing the unit infeed of the grinding wheel to the electrode also positively influences the reduction of the grinding temperature—*Θ_t_* and the increase of the relative grinding efficiency—*Q_v_*. For the individual power of the generator values, the reduction of the grinding temperature—*Θ_t_* was achieved amounting to respectively: for P1 (*U* = 100 V, *I* = 12 A)—temperature drop by approximately 5% in series II and series III; for P2 (*U* = 200 V, *I* = 25 A)—temperature drop by approximately 4% in series II and 6% in series III; for P3 (*U* = 300 V, *I* = 50 A)—temperature drop by approximately 4% for series II and 6% in series III. The increase in the relative grinding efficiency—*Q_v_* was observed: for P1 (*U* = 100 V, *I* = 12 A)—an increase of the *Q_v_* index by approximately 3% in series II and approximately 4% in series III, for P2 (*U* = 200 V, *I* = 25 A)—an increase of the *Q_v_* index by approximately 3% in series II and III, for P3 (*U* = 300 V, *I* = 50 A)—an increase of the *Q_v_* index by approximately 3% in series II and approximately 4% in series III. The lowest grinding temperature (*Θ_t_* = 537 °C) and the highest value of the relative grinding efficiency (*Q_v_* = 7.12 mm^3^/s) were obtained in the dressing conditions with the highest power of the generator P3 (*U* = 300 V, *I* = 50 A) and the largest unit infeed of the grinding wheel—*a_j_* = 0.015 mm.

#### 4.1.2. Dressing Tests of cBN Grinding Wheel

The results of the research concerning the assessment of the influence of the power of the electro-spark discharge generator and the unit infeed of the cBN grinding wheel to the stationary electrode on the grinding temperature—*Θ_t_* and the relative grinding efficiency—*Q_v_* are presented in Figure 8.

The analysis of the test results of dressing process of cBN grinding wheel using stationery electrode indicates a similar nature of the qualitative changes in the values of the grinding wheel machinability indexes *Θ_t_* and *Q_v_* compared to those obtained in the diamond grinding wheel dressing process (Figure 8). However, the differences concern quantitative changes in the values of these indicators. Increase of the power of the electro-spark discharge generator reduces the grinding temperature—*Θ_t_* by approximately 5–17% and the increase of the relative grinding efficiency—*Q_v_* by approximately 6–15%. Increase of the unit infeed of the grinding wheel in individual dressing series causes a reduction of the grinding temperature of the tester and an increase of the relative grinding efficiency—*Q_v_*. For the individual power of the generator values, the reduction of the grinding temperature—*Θ_t_*, was achieved amounting to respectively: for P1 (*U* = 100 V, *I* = 12 A)—temperature drop by approximately 3% in series II and 6% in series III, for P2 (*U* = 200 V, *I* = 25 A)—temperature drop by approximately 4% in series II and 6% in series III, for P3 (*U* = 300 V, *I* = 50 A)—temperature drop by approximately 4% in series II and 7% in series II and in series III. The increase of the relative grinding efficiency—*Q_v_* index was achieved amounting to respectively: for P1 (*U* = 100 V, *I* = 12 A)—an increase of the *Q_v_* index by approximately 3% in series II and 6% in series III; for P2 (*U* = 200 V, *I* = 25 A)—an increase of the *Q_v_* index by approximately 4% in series II and 6% in series III; for P3 (*U* = 300 V, *I* = 50 A)—an increase in the *Q_v_* index by approximately 7% in series II and series III. The lowest grinding temperature (*Θ_t_* = 609 °C) and the highest value of the highest value of the relative grinding efficiency (*Q_v_* = 6.63 mm^3^/s) were obtained under the dressing conditions with the highest power of the generator P3 (*U* = 300 V, *I* = 50 A) and the largest unit infeed of the grinding wheel—*a_j_* = 0.015 mm.

### 4.2. Electrodischarge Dressing of Grinding Wheels with a Rotating Electrode

The tests included three series of electrodischarge dressing tests of grinding wheels with diamond and cBN abrasives using rotating electrode, carried out with the following power settings of the electro-spark discharge generator: P1 (*U* = 100 V, I = 12 A); P2 (*U* = 200 V, *I* = 25 A); P3 (*U* = 300 V, *I* = 50 A). In each series of dressing tests, a constant value of the total infeed of the grinding wheel to the electrode (0.1 mm) was applied, divided into 10 unit infeeds, equal to 0.01 mm.

For the individual series of dressing tests, the dressing time was differentiated in the subsequent unit infeed of the grinding wheel to the working electrode, which was respectively: for series I—*t*_01_ = 100 s, for series II—*t*_02_ = 200 s, for series III—*t*_03_ = 300 s, for series IV—*t*_02_ = 600 s, for series V—*t*_03_ = 900 s. The scope of the research included also determining the effect of the power of the spark generator pulse generator and the unit infeed of the grinding wheel to the electrode on the grinding temperature—*Θ_t_* and relative grinding efficiency—*Q_v_*.

#### 4.2.1. Dressing Tests of Diamond Grinding Wheel

The results of the research concerning the the influence of the power of the electro-spark discharge generator and the unit infeed time of the diamond grinding wheel to the rotating electrode on the grinding temperature—*Θ_t_* and the relative grinding efficiency—*Q_v_* are presented in Figure 9.

The analysis of the test results of the dressing process of diamond grinding wheel using rotating electrode showed that: the increase in the power of the spark pulse generator has a positive effect on the reduction of the grinding temperature—*Θ_t_* and the increase of the relative grinding efficiency—*Q_v_* (Figure 9). For the applied power of the electro-spark discharge generator values, the percentage decrease of the grinding temperature *Θ_t_* for each series of grinding wheel dressing was respectively: for series I approximately 4% ÷ 10%, for series II approximately 10% ÷ 15%, for series III approximately 10 % ÷ 18%, for series IV approximately 6% ÷ 11%, for series V approximately 6% ÷ 14%. The increase in relative grinding efficiency—*Q_v_* for individual dressing series was respectively: for series I 5% ÷ 9%, for series II 4% ÷ 9%, for series III 4% ÷ 11%, for series IV 5% ÷ 9% and for series V 8% ÷ 13%. The tests carried out have shown that the dressing time in the subsequent unit infeed of the grinding wheel to the electrode has a diversified impact both on the grinding temperature—*Θ_t_* and the relative grinding efficiency—*Q_v_*. It was found that the lowest grinding temperature (*Θ_t_* = 486 °C) and the highest relative grinding efficiency (*Q_v_* = 7.44 mm^3^/s) were obtained in the second series of dressing, performed with the highest power of the electro-spark discharge generator P3 (*U* = 300 V, *I* = 50 A) and the dressing time of the grinding wheel *t*_02_ = 200 s.

#### 4.2.2. Dressing Tests of cBN Grinding Wheel

The results of the research concerning the the influence of the power of the electro-spark discharge generator and the unit infeed time of the cBN grinding wheel to the rotating electrode on the grinding temperature—*Θ_t_* and the relative grinding efficiency—*Q_v_* are presented in Figure 10.

The analysis of the test results for the dressing process of the cBN grinding wheel using rotating electrode indicates a similar nature of changes in the qualitative indicators of the machinability assessment of this grinding wheel—*Θ_t_* and *Q_v_* (Figure 10). However, the differences concern quantitative changes in the values of these indicators. It has been shown that: the increase in the power of the spark pulse generator has a positive effect on the reduction of the grinding temperature—*Θ_t_* and the increase of the relative grinding efficiency—*Q_v_*. The percentage decrease of the grinding temperature—*Θ_t_* for individual dressing series was as follows: for series I approximately 7% ÷ 14%, for series II approximately 5% ÷ 13%, for series III approximately 7% ÷ 15, for series IV approximately 6% ÷ 14 % and for series V approximately 6% ÷ 14%, while the increase in the relative grinding efficiency—*Q_v_* was respectively: for series I—8% ÷ 14%, for series II—5% ÷ 11%, for series III—7% ÷ 12%, for series IV—5% ÷ 12% and for series V—11% ÷ 15%. It has also been shown that the time of dressing in the subsequent unit infeed of the grinding wheel to the working electrode has a diversified influence on the grinding temperature—*Θ_t_* and relative grinding efficiency—*Q_v_*. It was found that the favorable dressing time in the subsequent unit feeds of the grinding wheel to the electrode was in the range *t_02_* = 200 s (series II). The lowest grinding temperature (*Θ_t_* = 572 °C) and the highest value of the relative grinding efficiency (*Q_v_* = 7.44 mm^3^/s) were obtained in the dressing conditions with the power of the electro-spark discharge generator P3 (*U* = 300 V, *I* = 50 A) and dressing time for series II (*t_02_* = 200 s).

### 4.3. Electroerodischarge Dressing of Grinding Wheels with a Segmental Electrode

The tests included three series of electrodischarge dressing tests of grinding wheels with diamond and cBN abrasives using segmental electrode, carried out with the following power settings of the electro-spark discharge generator: P1 (*U* = 100 V, I = 12 A)**;** P2 (*U* = 200 V, *I* = 25 A)**;** P3 (*U* = 300 V, *I* = 50 A).

In the individual dressing series, the longitudinal feed speed of the segmental electrode—*v_fe_* was changed, amounting to each series: for series I—0.25 m/min, for series II—0.5 m/min, for series III—0.75 m/min. The value of the unit infeed of the grinding wheel to the electrode was 0.01 mm, while the dressing time in the following unit infeeds was 60 s.

#### 4.3.1. Dressing Tests of Diamond Grinding Wheel

The results of the research concerning the influence of the power of the electro-spark discharge generator and the longitudinal feed of the electrode of the diamond grinding wheel to the segmental electrode on the grinding temperature—*Θ_t_* and the relative grinding efficiency—*Q_v_* are presented in Figure 11.

The analysis of the obtained results of the dressing process of diamond grinding wheel using segmental electrode shows the different nature of the influence of the power of the electro-spark discharge generator and longitudinal feed of the electrode on the grinding temperature—*Θ_t_* and the relative grinding efficiency—*Q_v_* (Figure 11). The lowest temperature of grinding the tester (*Θ_t_* = 496 °C) and the most favorable value of the relative grinding efficiency (*Q_v_* = 7.19 mm^3^/s) were obtained under dressing conditions with the middle value of the generator power of the generator P2 (*U* = 200 V, *I* = 25 A) and the smallest longitudinal feed of the electrode *v_fe_* = 0.25 m/min. It has been shown that increasing the longitudinal feed of the electrode and the power of the generator adversely affects on the increase of the grinding temperature—*Q_v_* and the deterioration of the relative grinding efficiency—*Q_v_*. Increasing the longitudinal infeed of the segmental electrode to the value of *v_fe_* = 0.5 m/min increases the grinding temperature—*Q_v_*, amounting to: approximately 7% for the power P1 (*U* = 100 V, *I* = 12 A), approximately 15% for the power P2 (*U* = 100 V, *I* = 12 A) and approximately 5% for the power P3 (U = 300 V, I = 50 A) and reduction of relative grinding efficiency—*Q_v_* amounting to: approximately 5% for P1 power (*U* = 100 V, *I* = 12 A), approximately 6% for the power P2 (*U* = 200 V, *I* = 25 A) and approximately 5% for the power P3 (*U* = 300 V, *I* = 50 A). A further increase in the longitudinal feed of the electrode to the value of *v_fe_* = 0.75 m/min also causes an increase of the grinding temperature—*Θ**_t_*, by approximately 15÷27%, and a reduction of the relative grinding efficiency—*Q_v_*, by approximately 6÷10%.

#### 4.3.2. Dressing Tests of cBN Grinding Wheel

The results of the research concerning the influence of the power of the electro-spark discharge generator and the longitudinal feed of the electrode of the cBN grinding wheel to the segmental electrode on the grinding temperature—*Θ_t_* and the relative grinding efficiency—*Q_v_* are presented in Figure 12.

The analysis of the test results of the dressing process of cBN grinding wheel using segmental electrode indicates a similar nature of the qualitative changes in the machinability assessment indexes of this grinding wheel—*Θ_t_* and *Q_v_*, to those obtained for the diamond grinding wheel dressing process. The lowest grinding temperature (*Θ_t_* = 605 °C) and the most favorable value of the relative grinding efficiency (*Q_v_* = 6.61 mm^3^/s) were obtained under dressing conditions with the middle value of the power of the generator P2 (*U* = 200 V, *I* = 25 A) and the lowest longitudinal feed of the electrode (*v_fe_* = 0.25 m/min). It has been shown that increasing the longitudinal feed of the electrode and generator power of the generator adversely affects on the increase of the grinding temperature—*Θ_t_* and the deterioration of the relative grinding efficiency—*Q_v_*. Increasing the longitudinal feed of the electrode to *v_fe_* = 0.5 m/min increases the grinding temperature—*Θ_t_*, to approximately 10% for P1 power (*U* = 100 V, *I* = 12 A), approximately 9% for P2 (*U* = 200 V, *I* = 25 A) and approximately 10% for power P3 (*U* = 300 V, *I* = 50 A) and reduction of relative grinding efficiency—*Q_v_* amounting to: approximately 10% for power P1 (*U* = 100 V, *I* = 12 A), approximately 8% for power P2 (*U* = 200 V, *I* = 25 A) and approximately 10% for power P3 (*U* = 300 V, *I* = 50 A). Further increasing the longitudinal feed of the electrode to the value of *v_fe_* = 0.75 m/min also causes an increase in the grinding temperature—*Θ_t_* amounting to 7% ÷ 12% and a decrease of the relative grinding efficiency—*Q_v_* amounting to approximately 6% ÷ 11%.

## 5. Conclusions

The conducted research confirmed the usefulness of the methods of electrodischarge dressing of superhard grinding wheels with a stationary, rotating and segment electrode on shaping their cutting abilities.

Objective evaluation of the cutting properties of superhard grinding wheels in the processes of electrodischarge dressing with a stationary, rotating and segment electrode is provided by the two-parameter method of grinding of the external tester, which is determined by the grinding temperature—*Θ_t_* and the relative grinding efficiency—*Q_v_*. The parameters of this method: *Θ_t_* and *Q_v_* show high sensitivity to changes in the conditions of electrodischarge dressing of superhard grinding wheels.

It has been shown that in the developed methods of electrodischarge dressing of superhard grinding wheels with a stationary, rotating and segment electrode, the power of the electro-spark discharge generator and the technological parameters (i.e., unit infeed of grinding wheel to electrode—*a_j_*, tester grinding time—*t*, longitudinal feed speed of the electrode—*v_fe_*) of these dressing methods have a significant impact on the shaping of the cutting ability of superhard grinding wheels. The increase in the power of the electro-spark discharge generator has a positive effect on the improvement of the machining index values *Θ_t_* and *Q_v_*.

Among compared electrodischarge dressing methods of superhard grinding wheels, the most favorable values of the machining index values *Θ_t_* and *Q_v_* were obtained for grinding wheels dressing with the rotating electrode, under the conditions of using the highest power of the electro-spark discharge generator P3 (*U* = 300 V, *I* = 50 A). Comparable values of the machining index values *Θ_t_* and *Q_v_* were also obtained for the dressing process using segmental electrodes under the conditions of the electro-spark discharge generator P2 (*U* = 200 V, *I* = 25 A).

The positive results of the research justify the continuation of work on electrodischarge dressing methods of superhard grinding wheels. The planned scope of research work in this area will include, among others:optimization of electrical parameters and technological conditions for the implementation of electrodischarge dressing of superhard grinding wheels with a stationary, rotating and segment electrode,improvement of design solutions for electrodischarge dressing of superhard grinding wheels with a rotating and segment electrode and their application in the industry,studies of selected physical phenomena occurring in the processes of dressing of superhard grinding wheels using these methods, in particular: evaluation of the micro- and nanogeometrical structure of the cutting surface of the grinding wheel, the impact of heat flux on the cutting properties of diamond and cBN grits.

Elaborated method of electrodischarge dressing of superhard grinding wheels with segmental electrode received polish patent no PL 190236 [42].

## Figures and Tables

**Figure 1 materials-14-06773-f001:**
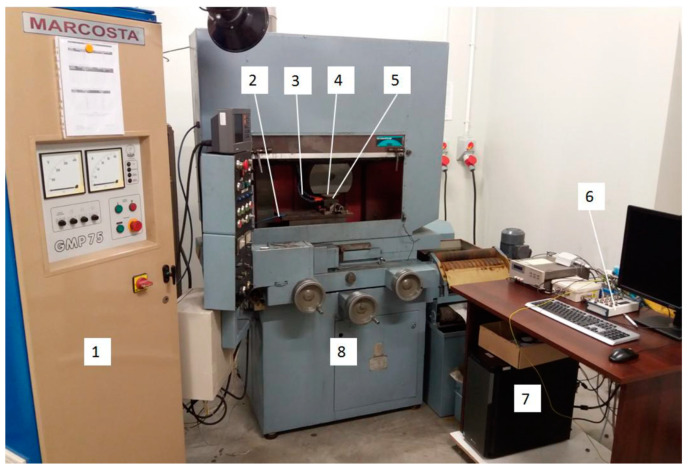
Test stand: 1—electro-spark discharge generator, 2—electric voltage supply to the working electrode, 3—dielectric liquid nozzle, 4—grinding wheel, 5—working electrode, 6—card terminal, 7—PC computer, 8—ECBT8 grinder.

**Figure 2 materials-14-06773-f002:**
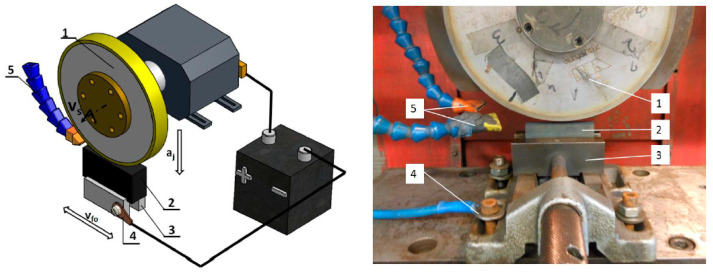
Diagram and view of the system for electrodischarge dressing of superhard grinding wheels with a stationary electrode: 1—grinding wheel, 2—stationary electrode, 3—machine vice jaws, 4—supplying electrical voltage from the electro-spark discharge generator, 5—dielectric supply nozzles.

**Figure 3 materials-14-06773-f003:**
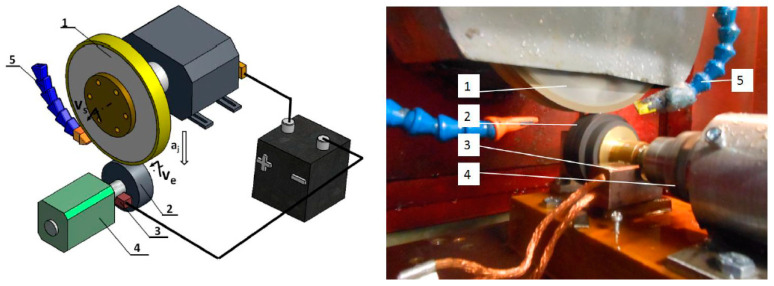
Diagram and view of the system for electrodischarge dressing of superhard grinding wheels with a rotating electrode: 1—grinding wheel, 2—rotating electrode, 3—copper graphite brush supplying voltage from the generator to the electrode, 4—pneumatic motor, 5—dielectric supply nozzles.

**Figure 4 materials-14-06773-f004:**
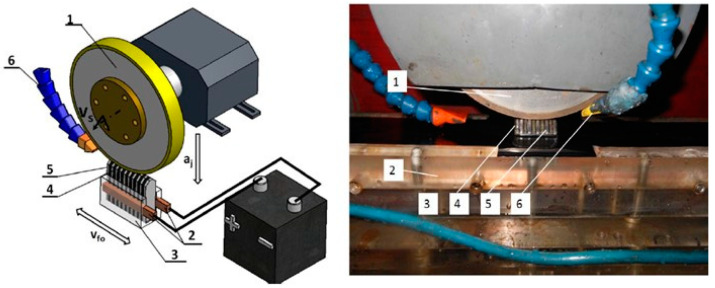
Diagram and view of the system for electrodischarge dressing of superhard grinding wheels with a segmental electrode: 1—grinding wheel, 2—copper graphite brush supplying voltage from the generator to the electrode, 3—vice, 4,5—segmental electrode, 6—dielectric supply nozzles.

**Figure 5 materials-14-06773-f005:**
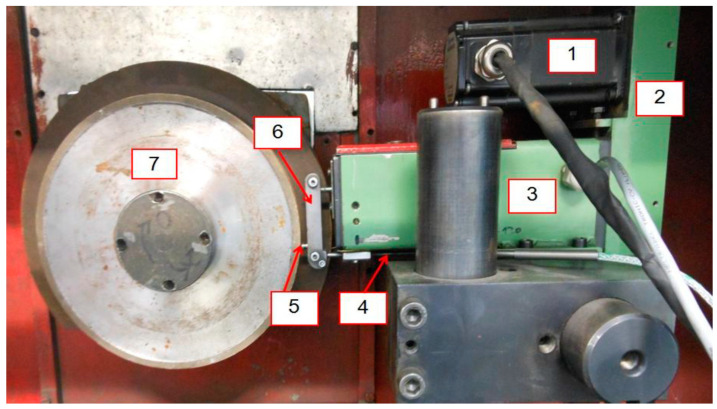
A measuring device for assessing the grinding ability of grinding wheels: 1—stepper motor, 2—toothed gear, 3—sample feed mechanism with a force sensor, 4—tester guide, 5—tester with thermocouple, 6—mounting clamp, 7—grinding wheel.

**Figure 6 materials-14-06773-f006:**
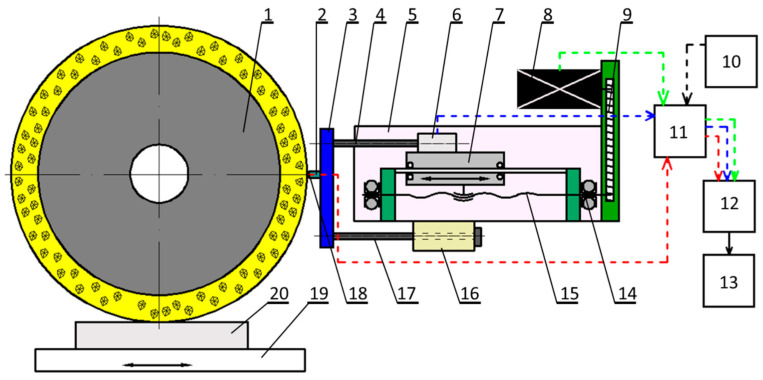
Scheme of the apparatus for assessing the cutting ability of grinding wheels: 1—grinding wheel, 2—tester (titanium alloy sample with thermocouple), 3—mounting bracket, 4—guide, 5—device body, 6—force gauge, 7—slide, 8—stepper motor, 9—toothed gear, 10—power supply, 11—apparatus controller, 12—PC card, 13—computer, 14—bearing, 15—helical-ball gear, 16—auxiliary clamp, 17—guide rail, 18—sheathed thermocouple, 19—grinder table, 20—workpiece.

**Figure 7 materials-14-06773-f007:**
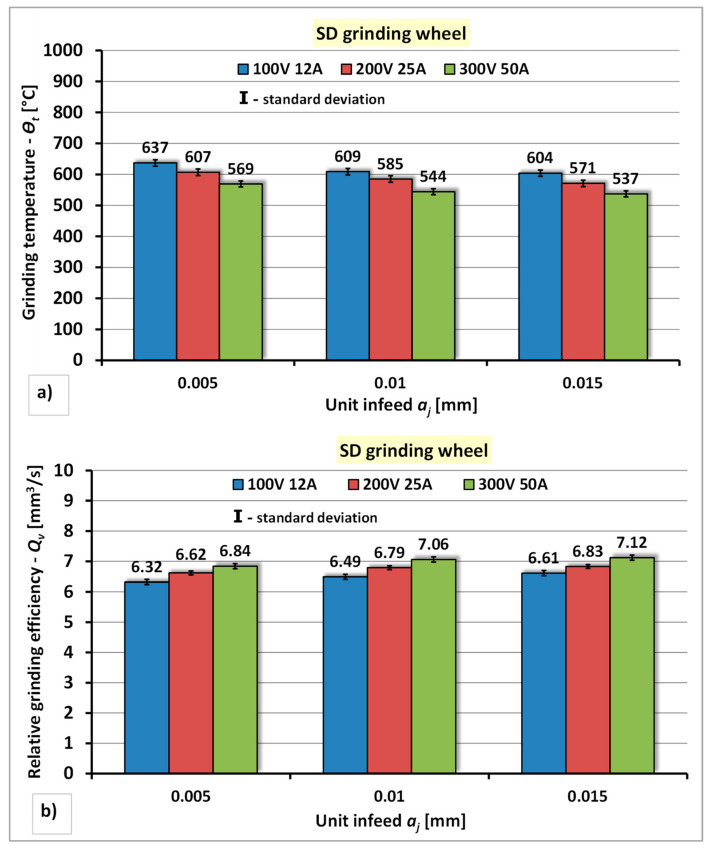
The effect of the generator power and unit infeed during electrodischarge dressing of a diamond grinding wheel using a stationary electrode: (**a**) On the grinding temperature—*Θ_t_*; (**b**) On the relative grinding efficiency—*Q_v_*.

**Figure 8 materials-14-06773-f008:**
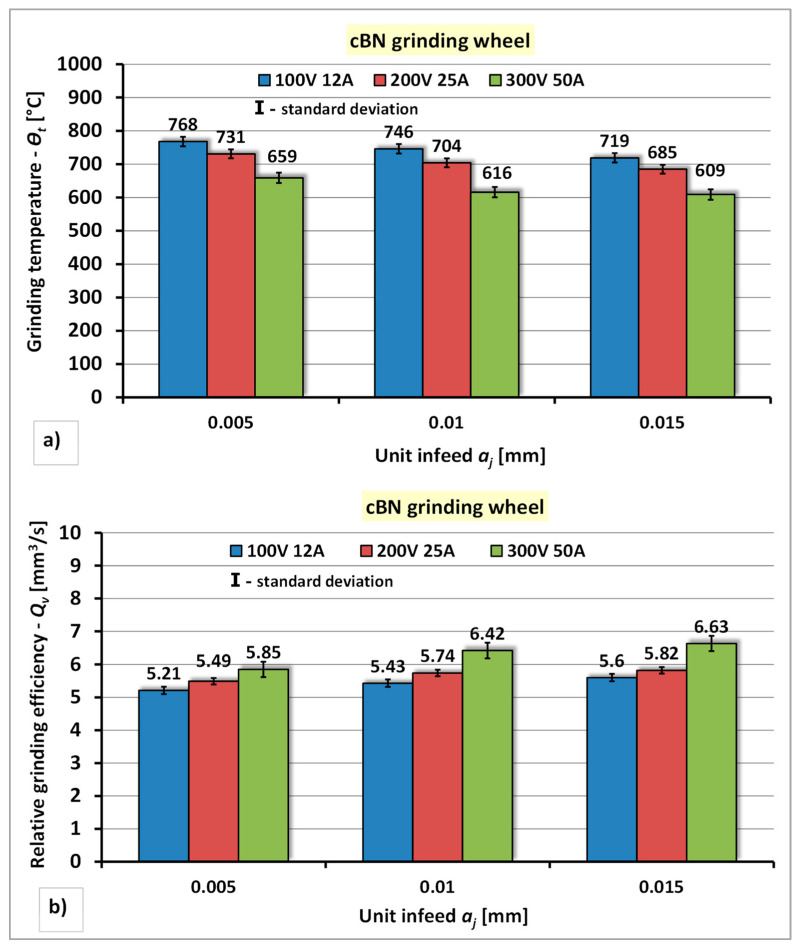
The effect of the generator power and unit infeed during electrodischarge dressing of a cBN grinding wheel using a stationary electrode: (**a**) On the grinding temperature—*Θ_t_*; (**b**) On the relative grinding efficiency—*Q_v_*.

**Figure 9 materials-14-06773-f009:**
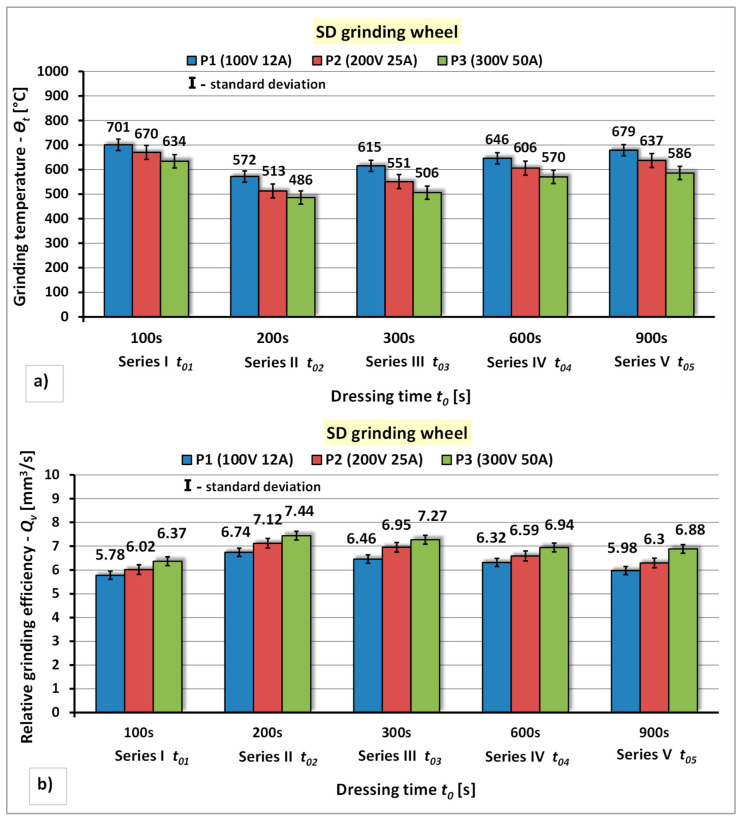
The effect of the generator power and unit infeed time during electrodischarge dressing of a diamond grinding wheel using a rotating electrode: (**a**) On the grinding temperature—*Θ_t_*; (**b**) On the relative grinding efficiency—*Q_v_*.

**Figure 10 materials-14-06773-f010:**
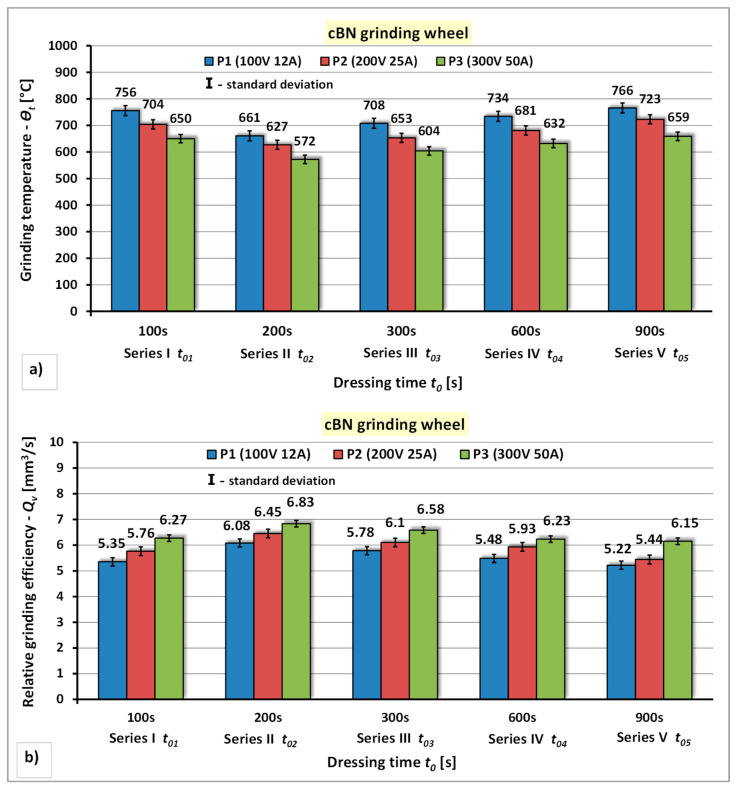
The effect of the generator power and unit infeed time during electrodischarge dressing of a cBN grinding wheel using a rotating electrode: (**a**) On the grinding temperature—*Θ_t_*; (**b**) On the relative grinding efficiency—*Q_v_*.

**Figure 11 materials-14-06773-f011:**
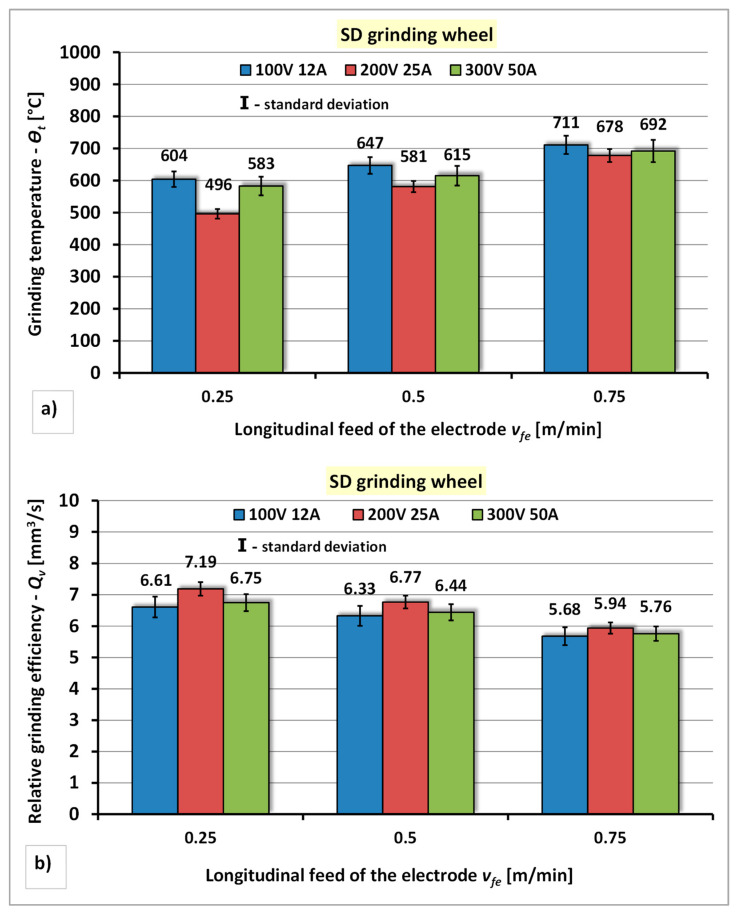
The effect of the generator power and longitudinal feed of the electrode during electrodischarge dressing of a diamond grinding wheel using a segmental electrode: (**a**) On the grinding temperature—*Θ_t_*; (**b**) On the relative grinding efficiency—*Q_v_*.

**Figure 12 materials-14-06773-f012:**
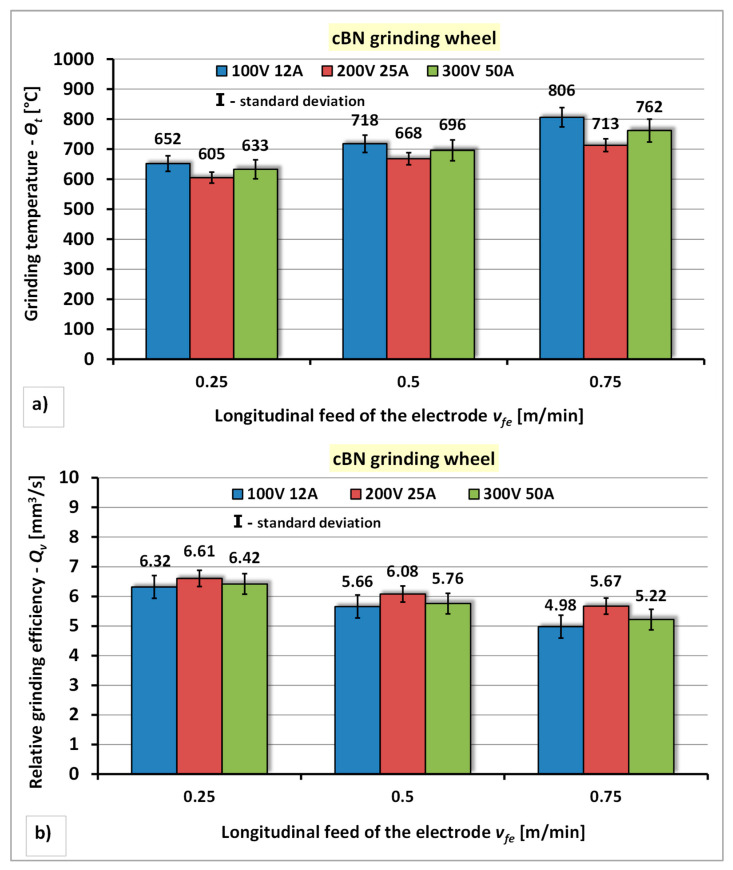
The effect of the generator power and longitudinal feed of the electrode during electrodischarge dressing of a cBN grinding wheel using a segmental electrode: (**a**) On the grinding temperature—*Θ_t_*; (**b**) On the relative grinding efficiency—*Q_v_*.

**Table 1 materials-14-06773-t001:** Characteristics of grinding wheels and machine tools used for the tests.

Technical Designation	S1010 200 × 20 × 5 × 32 SD 125/100 M100 S1010 200 × 20 × 5 × 32 cBN 125/100 M75
Manufacturer	“VIS” S.A. (Warszawa, Poland)
Type of grinding wheel	Straight grinding wheel
Dimensions	External diameter *D* = 200 mm, inner diameter *d* = 32 mm, abrasive layer width *w* = 20 mm, thickness of abrasive layer *x* = 5 mm
Type of abrasive	Monocrystalline synthetic diamond (SD) Cubic boron nitride (cBN)
Grit size	125, FEPA, PN-85/M-59108 [µm]
Bond	Metal
Grit concentration	100 (SD), US Standard ASTM E11 [mesh] 75 (cBN), US Standard ASTM E11 [mesh]
Grinding machine	ECBT8, TOS Hostivař (CETOS); Czech Republic

## Data Availability

Data is contained within the article.

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
