# Peer review of "Electrodischarge Methods of Shaping the Cutting Ability of Superhard Grinding Wheels"

_materials, 2021, doi:10.3390/ma14226773_

Round 1

Reviewer 1 Report

Comments and Suggestions are provided in the attached PDF.

Reviewer 2 Report

Dear authors,

your paper is very interesting and up todate.  I have few comments:

  1. I suggest you to change the word "shaping" into assesing, or evaluate, aspecially in the title of the paper.
  2. Sentence (line 69, 70, 71,72) needs reconsideration (less nuisance for the natural environment, and the environment ???)
  3. Could you explain how did you measure or better to say controlle the force with Kistler dynamometer 9257B?
  4. What for did you used Profilometer you mentioned in line 90?
  5. At least, you have done a lot of experiments and given quantitative explanations, which are also visible from the diagram. Can you go a little deeper into the problem and explain from the physical side, or from the material side, why some parameter affects your output quantities.
  6. Your statement that segmental electrode does not need an insulation shoul be explained, why is that so?

Reviewer 3 Report

  1. The Grammar of this manuscript needs to be carefully revised. It is really hard to read through the manuscript. Double check punctuation marks, word capitalization, and tense throughout the entire paper.
  2. It is very hard to find the motivation of this manuscript and it is a very brief introduction about what the author will do in this manuscript.

  3. Include scope for further study in the conclusion part.

Round 2

Reviewer 1 Report

Despite some positive modifications that were introduced, this version still contains some flaws.

1) In my opinion the new title (“Electrodischarge Methods of Assessing the Cutting Ability of Superhard Grinding Wheels”) does not correctly describe the scope of this research. In fact, “electrodischarge methods” are used for “dressing” the tools, and not for assessing/evaluating/measuring the cutting ability of the tools. The first title (“Electrodischarge Methods of Shaping the Cutting Ability of Superhard Grinding Wheels”) would be preferable, considering that “shaping” is a term that means “influencing”.

2) As mentioned in my previous report, there are too many unnecessary repetitions of words. Just as one example, the following paragraph (lines 16 – 19) could be written in a much simpler way: … “The cutting ability of superhard grinding wheels are assessed using an external tester made of titanium alloy 17 Ti-6Al-4V to determine the grinding temperature and the relative volumetric grinding efficiency of the tool.

Along the text (see lines: 48, 49, 80, 81, 168, 217, 217, 230, 231, 236, 239, 240, 243, 244, 245, 247, 250, 251, 252, also at y-axes of Fig.7, 260, 261, 267, 268, 271, 272, also at y-axes of Fig.8, 278, 279, 281, 282, 283, 292, 293, 308, 309, 314, 318, 321, 322, 324, 331, 332, 333, 334, also at y-axes of Fig.9, 344, 345, 351, 352, also at y-axes of Fig.10, 353, 356, 360, 263, 364, 380, 384, 388, 394, also at y-axes of Fig.11, 397, 402, 403, 409, 413, also at y-axes of Fig.12, 418, 423, 425, 431, 440, 441, 501, 502) why the authors always use “grinding temperature of the tester” and “grinding efficiency of the tester”?

In fact, the objective of the external tester is to measure “temperature” of the grinding process and to determine “volumetric grinding efficiency” (not “of the tester”, but “of the grinding process”). Therefore, all those unnecessary repetitions of “of the tester” can be deleted.

3) In the revised version, the authors have introduced a fragment of text where they present the definition of “relative grinding efficiency” (see lines 81 – 86). It would be more adequate to provide the meaning of the symbols Vw and t immediately after Eq.(1) and not only at the list of Nomenclature (lines 505 and 506). Also, it is necessary that the authors describe the experimental methodology that was used to determine the values of Vw (volume of grinded tester).
